# Mitochondria in Cell-Based Therapy for Stroke

**DOI:** 10.3390/antiox12010178

**Published:** 2023-01-12

**Authors:** Molly Monsour, Jonah Gordon, Gavin Lockard, Adam Alayli, Cesar V. Borlongan

**Affiliations:** 1University of South Florida Morsani College of Medicine, Tampa, FL 33602, USA; 2Center of Excellence for Aging and Brain Repair, Department of Neurosurgery and Brain Repair, University of South Florida Morsani College of Medicine, Tampa, FL 33612, USA

**Keywords:** stroke, stem cell, mitochondria, oxidation, neuroinflammation, reactive oxygen species

## Abstract

Despite a relatively developed understanding of the pathophysiology underlying primary and secondary mechanisms of cell death after ischemic injury, there are few established treatments to improve stroke prognoses. A major contributor to secondary cell death is mitochondrial dysfunction. Recent advancements in cell-based therapies suggest that stem cells may be revolutionary for treating stroke, and the reestablishment of mitochondrial integrity may underlie these therapeutic benefits. In fact, functioning mitochondria are imperative for reducing oxidative damage and neuroinflammation following stroke and reperfusion injury. In this review, we will discuss the role of mitochondria in establishing the anti-oxidative effects of stem cell therapies for stroke.

## 1. Introduction

The central nervous system is a highly active, energy intensive organ system that relies on careful homeostasis to maintain smooth function. The high energy demand places mitochondria in the forefront as the pivotal organelle maintaining and supplying neuronal energy demands [1]. Mitochondria are multi-functional organelles found in eukaryotic cells, with roles, such as energy production, regulation of apoptosis, buffering intracellular calcium, and the development of reactive oxygen species [2,3,4]. They are integral to functions within the central nervous system because they produce a majority of the energy needed for membrane ATPases, the influx and efflux of neurotransmitters, and the formation of new neural circuits [1,5].

Mitochondria exhibit an adaptive response to the fluctuating needs of their host cell in an effort to maintain bioenergetic and oxidative homeostasis [6,7]. In large, complex cells such as neurons, mitochondrial distribution plays a critical role in supplying the cell with needed energy [8,9]. The organelle may even move around the cell in response to metabolic demand. For example, certain mitochondria can move along axons using kinesin or dynein motors to reach areas with higher energy needs [10]. Other mitochondria are anchored into the membrane and remain stationary to supply a continuous source of energy to a local structure within the cell such as in dendrites [11]. Mitochondria may also fuse or undergo fission in order to respond to fluctuating energy demands. For example, mitochondrial fission allows the organelles to travel into growing spines and dendrites to promote neurogenesis and plasticity while blocking mitochondrial fission that results in neuronal degeneration and disruption of neuronal morphology [12,13].

Although neurons are typically lifelong cells and do not regenerate, mitochondria experience regular turnover to remove damaged organelles and minimize the unintentional release of proapoptotic signals and accumulation of reactive oxygen species [14]. This process is called mitophagy and occurs in cells all over the body. In the setting of neurodegenerative disease, mitophagy decreases and injured mitochondria accumulate. Normally, upon damage to mitochondria, there is a subsequent elevation of PTEN-induced kinase 1 (PINK1) in the mitochondrial outer membrane, which phosphorylates, and thus activates E3 ubiquitin ligase Parkin [15,16]. In one mechanism, Parkin ubiquitinates and thus eliminates mitochondrial fusion proteins mitofusin1 and mitofusin2, resulting in mitochondrial fission, separating diseased mitochondrial components [17]. Additionally, Parkin ubiquitinates mitochondrial surface proteins; the ubiquitin moieties attract a battery of autophagy receptors, including optineurin and nuclear dot protein 52, among others. The autophagy receptors bind to LC3, a protein embedded in the autophagosome, thus connecting the two organelles, and allowing for mitophagy [18]. There is a careful balance of mitophagy that must be struck; too much can be detrimental to cell life and too little can be factorial in the etiology of neurodegenerative disease [19]. Defects in mitophagy are implicated in the development of neurodegenerative diseases, such as Alzheimer’s and Parkinson’s disease, and restoring this essential function may offer a therapeutic target for these diseases [20].

## 2. Mitochondrial Impairment in the Oxidative Stress following Stroke and Reperfusion Injury

Ischemic stroke is defined as a sustained lack of blood flow to an area of the brain resulting in local inflammation, oxidative stress, and cellular death. While the primary mechanisms of cell death relate to ischemic injury, mitochondrial damage is a primary component of secondary cell death, contributing to excitotoxicity, oxidative stress, free radical accumulation, impaired neurogenesis, angiogenesis, vasculogenesis, and inflammation [21,22]. Within the mitochondria, the Krebs cycle transfers energy from glycolytic molecules to electron carriers, propagating the oxidative phosphorylation pathway of maximal ATP production [23]. Mitochondria are vital to energy production via this oxidative phosphorylation; however, oxidative phosphorylation also notoriously results in free radical accumulation [24]. Thus, if malfunctioning, mitochondrial damage can result in reduced energy production and excessive accumulation of free radicals and oxidative stress following ischemic injury [25]. Furthermore, ischemic injury to mitochondria leads to their programmed death and release of cytochrome C. Cytochrome C then perpetuates neuronal death via apoptosis, furthering the release of ROS [26]. As oxidative phosphorylation requires oxygen, and ischemic stroke is defined by a lack of oxygen to neurons, it is incredibly logical to target underlying mitochondrial damage to ameliorate further cell death secondary to oxidative stress. Furthermore, reperfusion following ischemic stroke generates a large number of ROS, contributing to oxidative stress [27].

## 3. Repair of the Damaged Mitochondria in Stroke: Astrocytes-to Neurons Transfer of Mitochondria

As mitochondria are, by nature, functioning within eukaryotic cells, many groups have examined the role mitochondria play in the successful treatment of stroke with stem cells [28]. More mysterious, however, is whether the mitochondria mitigate ischemic damage via direct transfer or signaling molecules (Figure 1). Some research suggests stem cells physically transfer healthy mitochondria to deteriorating neuronal cells, similar to astrocytic aid in neuronal survival following stroke [29,30]. Others, however, perceptualize that direct mitochondrial energy metabolism within stem cells can modulate the stem cells’ differentiation, ageing, immune regulation, apoptosis, proliferation, migration, and chemotaxis [31].

The immediate impacts of mitochondrial repair are extended via downstream antioxidant effects. ROS inhibit the Nuclear factor erythroid 2-related factor 2 (Nrf2) anti-oxidant pathway. Thus, by blocking the accumulation of ROS via mitochondrial transfer, the Nrf2 pathway can be restored following stroke and implement its anti-inflammatory and anti-oxidant pathways. Regarding cell-based therapies, the downstream reduction in ROS can also amplify stem cell viability and function. For instance, upregulation of Nrf2 and downstream antioxidant genes such as HO-1 augments neurogenesis and increases NSC viability via decreased apoptosis [32,33]. Thus, successive cell-based therapies may be synergistically effective due to the downstream antioxidant effects of restoring mitochondrial viability with the first cell dose. Alternatively, pre-emptive treatment with antioxidant agents may lower the workload of stem cells and their mitochondrial progenies [26]. Thus, there is a promising tie between enhancing mitochondrial efficacy and mitigating oxidative stress following ischemic stroke. Ultimately, the profound impacts of restoring mitochondria functionality can contribute to neurological repair via its downstream anti-inflammatory and antioxidant impacts.

## 4. Stem Cell-Neural Cell Crosstalk: Rescue of Mitochondria by Stem Cells

Mitochondria play a mitigating role in neuroinflammation, such as in the setting of ischemic cerebrovascular accident. This is of particular interest with the use of stem cell therapy. Stem cells exert their effects on endogenous cells in a variety of ways; they can release molecules to communicate in a paracrine fashion [34], release exosomes [35], and even alter energetic efficacy, inflammation, and oxidative stress via mitochondrial adaptations. After treatment with human bone marrow endothelial progenitor cells, rat models of ischemic stroke had restored endothelial cell, pericyte, and astrocyte morphology. Upon closer analysis, mitochondrial morphology was restored in these cells as well, suggesting changes to mitochondrial integrity may be responsible for this therapy’s beneficial effect [36]. Regarding stem cell impacts on mitochondria, various theories have been proposed, such as direct mitochondrial transfer and mitochondrial metabolite transmission (Table 1 and Table 2).

Stem cells can form intercellular bridges named tunneling nanotubes [45], or directly fuse; these latter two mechanisms allow the exchange of cellular contents including organelles such as mitochondria [46]. The exchange of healthy mitochondria into an ischemic cell is very promising. Indeed, Babenko et al. found that mitochondrial transfer from multipotent mesenchymal stem cells (MMSC) to astrocytes is more robust in the setting of elevated reactive oxygen species (ROS) secondary to ischemic insult, compared to normal conditions. Mitochondrial Rho-GTPase 1 (Miro1) is a calcium-sensitive adaptor protein, assisting in the intracellular and intercellular transport of neuronal mitochondria [47], and when Miro1 is overexpressed in MMSCs, the mitochondrial donation to astrocytes is markedly increased [37]. Additionally, rats who underwent the middle cerebral artery occlusion (MCAO) stroke model and were then treated with MMSC-Miro1 had improved neurological function compared to MMSC therapy alone. Similarly, Liu et al. found that grafted MSCs in rats who underwent MCAO/reperfusion offered mitochondria to injured endothelial cells, rescuing mitochondrial activity and improving angiogenesis and neurologic function [38].

Direct implantation of mitochondria may surpass the cell-exudation step and provide equally beneficial results. In an OGD brain endothelial model, human endothelial progenitor cell-derived extracellular mitochondria promoted angiogenesis, blood brain barrier (BBB) impermeability, and increased ATP [44]. These results are incredibly pertinent to recent theories regarding BBB permeability as a major contributor to secondary cell death in stroke due to peripheral inflammatory and ROS influx [48,49]. In vivo, human umbilical cord-MSC-derived mitochondria also improved ischemic injury in rats, with treated rats showing decreased apoptosis, astrogliosis, neuroinflammation, and infarct size. Furthermore, some motor functional recovery was notable [39]. On a similar note, BM-MSC-derived mitochondria given to mice models of stroke reduced ROS, improves memory, enhanced energy efficacy, and increases synaptic marker expression. Interestingly, this study used an intranasal route for mitochondrial treatment, introducing a less invasive and possibly more direct mechanism of cell-based therapy administration [41]. A more direct route to the ischemic region would likely amplify cell survival and induce a more potent anti-inflammatory and antioxidant effect. Further studies should examine whether this administration method is superior to the more standard intravenous or intraarterial routes of administration. Whether exuded by stem cells themselves or extracted and implanted artificially, mitochondrial transfer demonstrates a clear benefit to reducing secondary mechanisms of cell death following stroke.

Of additional interest are the potential antioxidant roles of mitochondria. While direct mitochondrial transfer was not noted in the following studies, variations in mitochondrial function may lead to mitochondrial metabolite transmission and reduced ischemic damage. Coenzyme Q10 (CoQ10) is a component of the electron transport chain in mitochondria, and when applied to neural stem cells in the setting of hypoxia-reperfusion, defends them from injury. CoQ10 attenuates free radical formation and increases the expression of antiapoptotic phosphorylated Akt (pAkt), phosphorylated glycogen synthase kinase 3-β (pGSK3-β), and B-cell lymphoma 2 (Bcl-2), while simultaneously decreasing proapoptotic cleaved caspase-3 [42]. DJ-1 is another molecule associated with antioxidant activity. DJ-1 promotes nuclear factor erythroid 2–related factor 2 (Nrf2) [50], which is a master switch for antioxidant genes, and decreases the expression of proapoptotic Bax [51]. In the oxygen-glucose deprivation (OGD) in vitro stroke model, it was observed that DJ-1 located to intact mitochondria, and when anti-DJ-1 antibody was administered, glutathione concentrations increased, and mitochondrial activity diminished [43].

Intriguingly, pre-conditioning stem cells may further exacerbate the beneficial impact of mitochondria in cell-based therapies for stroke. In one study, growing mesenchymal stem cells (MSC) under a metabolic switching paradigm (three days in a galactose medium and three days in a glucose medium) resulted in enhanced therapeutic effects of SCs in an in vitro OGD model of stroke. The MSCs grown under a metabolic switching paradigm generated more ATP, shifted their metabolism to favor oxidative phosphorylation, had a higher basal energy production, had a greater spare respiratory capacity, and leaked less hydrogen ions. Thus, these cells demonstrated greater efficiency during energy production and a superior ability to respond to stressful conditions (i.e., ischemic energy) compared to the MSCs cultured with a standard glucose medium. When applied to the in vitro stroke model, the switched MSCs showed greater cell viability, enhanced metabolism, reduced ROS from stem cell mitochondria, and elevated mitochondrial ATP production by the MSCs compared to the standardly cultured cells [25,30]. Since the only difference between the MSC groups involved metabolism, which is predominantly moderated by mitochondria, it is theorized that the superior efficacy of metabolically switched MSCs is due to the generation of “super mitochondria”. It is plausible that the MSCs transferred these mitochondria to the damaged cells, allowing for greater recovery of ATP and metabolic recovery. Other groups have shown that, in vitro, co culture of PGD neurons with astrocyte conditioned media (including mitochondrial particles), restores neuronal viability and ATP production. Using fluorescent staining, the transfer of astrocytic mitochondria to ischemically stressed neurons was confirmed. In vivo, these results held true, showing that astrocytes use a CD38/cyclic ADP ribose signaling pathway to facilitate mitochondrial transfer to neurons following stroke in mice models [29]. Other studies have elucidated the amplification of stem cells’ antioxidant properties following pre-conditioning as well. In rats treated with ischemic-hypoxic pre-conditioned olfactory mucosa MSCs, mitochondrial function was preserved, and ROS levels were significantly reduced [40].

Although the antioxidant and antiapoptotic properties of stem cells and mitochondria are exciting fields to investigate further, the deleterious effects of ROS on cells are indisputable. Prakash et al. investigated the effects of hydrogen peroxide on human dental pulp and MSC. Hydrogen peroxide triggered oxidative stress, prompting autophagy and mitophagy [19]. As discussed earlier, mitophagy is beneficial in clearing injured mitochondria, but there is a health balance that must be maintained. Ultimately, the ideal cell-based therapy would be able to withstand these toxic impacts of ROS, while simultaneously reducing ROS induced secondary cell death. Ongoing studies on mechanisms of mitochondrial repair following cell-based therapies show promise for uncovering this lucrative therapeutic goal.

## 5. Non-Cell-Based Approaches to Mitochondrial Repair in Stroke

Recent advancements in cell-based therapies encourage researchers to explore the processes of mitochondrial transfer from stem cells, however, other approaches to restoring mitochondrial integrity following stroke may involve pharmaceutical or life style changes [52]. Resveratrol, a SIRT1 activator, for instance, reduces ROS and ultimately improves mitochondrial glucose utilization and restoration [53,54]. Considering 54 studies of stroke rodent models, resveratrol decreased infarct volumes and improved neurobehavioral scores, likely via its enhancement of mitochondrial function [55]. Moreover, in an effort to reduce oxidative stress by restoring mitochondrial function, methylene blue can enhance the transfer of electrons along the oxidative phosphorylation pathway to reduce ROS and increase ATP production efficiency [56]. In rats, methylene blue treatment normalized blood flow, decreased ischemic burden, and enhanced rodent functionality [57].

Oxidative burden may be effectively reduced by pharmaceuticals following stroke, but simple dietary additives such as ubiquinone, N-acetylcysteine, and/or anti-oxidant vitamins (i.e., C and E) may also reduce ROS following ischemic stroke [52]. Diet changes may also harbor potential to strengthen mitochondrial integrity following stroke. As previously discussed, stem cells cultured under metabolic switching conditions showed increased mitochondrial ATP production and decreased mitochondrial ROS mRNA when used to treat OGD neurons. These stem cells amplified oxidative phosphorylation over glycolytic metabolism to enhance the efficiency of energy metabolism. Metabolic switching in vitro may be mirrored by a ketogenic diet, which similarly minimizes glucose availability and could shift cells to generate more powerful mitochondria [25,30]. Yet another lifestyle adjustment includes exercise, as increased physical activity may improve mitochondrial function by enhancing oxidative phosphorylation and reducing oxidative stress [52]. Ultimately, while cell-based therapies are an exciting and rapidly developing field, alternative therapeutics may harbor benefits by similarly restoring mitochondrial integrity after stroke.

A number of clinical trials have been conducted with the goal of reducing oxidative stress following stroke [27]. Lipoic acid, involved in vitamin C and E recycling, is being tested in an ongoing clinical trial on reperfusion therapies for ischemic stroke in patients with diabetes (NCT 04041167). Salvianolic acid is also being studied as an antioxidant following stroke, as it is a potent oxidation, coagulation, and platelet aggregation inhibitor (NCT04931628). Edaravone, a ROS scavenger, has shown reduced mortality when administered with alteplase therapy or after endovascular revascularization [58,59]. Importantly, not all preclinical successes are translatable to clinical settings, however, and further investigation is needed [27,60,61,62,63,64,65,66].

## 6. Mitochondrial Repair in Other Disorders of the Central Nervous System

While the rescue of mitochondria has therapeutic promise in the treatment of stroke, mitochondrial damage plays a significant role in other degenerative CNS disorders, such as Alzheimer’s disease (AD), Parkinson’s disease (PD), and amyotrophic lateral sclerosis (ALS) [67]. As such, these conditions may also benefit from mitochondrial repair (Table 3).

Alzheimer’s disease is a neurodegenerative disorder often associated with the accumulation of amyloid-β plaques in the brain, although the true etiology of the disease is unclear and increasingly seems to be multifactorial in nature. The exact molecular cause of mitochondrial dysfunction in AD increasingly appears to be complex and not relegated to a single pathway. Early stages of the disease are characterized by degradation of mitochondrial function, such as a loss of calcium homeostasis, dysregulated neuronal apoptosis, and a significant increase in the abundance of reactive oxygen species [75,76]. The accumulation of (NH(2))-derived tau protein and amyloid-β plaques in mitochondria inhibits the function of the mitochondrial adenine nucleotide translocator-1, which suggests a potential mechanism for early mitochondrial dysfunction in AD [77]. Amyloid-β plaques also seem to inhibit mitochondrial transport, fission, and fusion, leading to neural degeneration and significantly decreased neuroplasticity [75]. Additionally, p9 mitochondrial tRNA hypermethylation seems to play a role in the pathogenesis of Alzheimer’s disease by impairing RNA stability, translational processes and disrupting downstream pathways within the mitochondria [78]. For example, impaired protein translation as a result of hypermethylated p9 may cause a disruption in the folding of proteins involved in the electron transport chain, signaling dynamics, and homeostasis. On a macro scale, astrocytes seem to increase their rate of transmitophagy in AD mouse models, suggesting yet another mechanism for neurodegeneration. This disrupts the function of affected astrocytes by increasing reactive oxygen species and altering the supportive relationship between astrocytes and neurons [79]. More research must be performed to assess the role of transmitophagy in the initiation of neurodegenerative disease, although this finding suggests that the factors at play are not limited to intracellular and intra-mitochondrial processes. In fact, mitochondrial transfer in mice models of AD demonstrates improved cognition, decreased neuronal loss and gliosis in the hippocampus, and improved mitochondrial functionality [72]. Mitochondrial integrity in AD may also be restored by inhibiting the voltage-dependent anion channel-1 (VDAC-1), a mitochondrial protein overexpressed in AD. In vitro and in vivo inhibition of VDAC-1 resulted in reduced neuronal apoptosis, decreased inflammation, and repaired metabolic function. Cognitive decline was also ameliorated in treated mice. While metabolic homeostasis was restored, likely by restoring normal mitochondrial protein function, the Amyloid-β and hyperphosphorylated tau burdens were not decreased [74]. This is of interest as it suggests mitochondrial dysfunction may be equally pertinent to AD pathology as the well-recognized plaque and tangle pathologies. Stem cell therapy for AD is a focus of current research and early studies show that the transplantation of neural stem cells from mouse embryos may rescue the spatial learning and memory function and enhance the long-term potentiation of APP/PS1 transgenic mice [80]. Human neural stem cells also display a similar benefit by enhancing synaptogenesis and offer the benefit of a lower chance of rejection after implantation into humans [81]. Furthermore, human embryonic stem cells transplanted into the basal forebrain of AD mouse models differentiated into mature cholinergic neurons and subsequently rescued the mice’s cognitive deficits and improved spatial learning [82].

The hallmark of Parkinson’s disease is the degeneration of dopaminergic neurons in the substantia nigra. Its pathogenesis is multifactorial, but many associated genes, such as SNCA, LRRK2, VPS35, PINK1, DJ-1, and Parkin have been identified and linked to mitochondrial dysfunction [83]. For example, mutations in the SNCA gene cause overexpression of α-synuclein, which can disrupt mitochondrial membrane tethers and result in loss of ATP production and calcium homeostasis [84]. Gains of function in LRRK2, another commonly mutated gene, play a role in neurotransmitter release and arrest mitochondrial fission within the substantia nigra [85]. Loss-of-function mutations in Parkin, the most common genetic cause of autosomal recessive PD, allow for the accumulation of damaged mitochondria by removing the ability to degrade and ubiquitin-tag mitochondrial proteins [86]. Loss of Parkin also leads to the accumulation of parkin interacting substrate (PARIS), which causes a decrease in mitochondrial biogenesis and quantity [87]. Most therapeutics for PD focus on alleviating symptoms and there is currently no effective cure for the disorder. Due to the major role of mitochondria in PD pathology, mitochondrial transfer may attenuate symptomology [88,89]. In 6-OHDA rat models of PD, mitochondria injected into the medial forebrain bundle improve mitochondrial function within substantia nigra neurons. These new mitochondria reduced dopaminergic cell death, improved cell functionality, and reduced oxidative DNA damage [69]. In mice, similar mitochondrial administration decreased ROS levels, cell apoptosis, and necrosis [71]. Stem cell therapy is a promising path because it may replace diseased cells and slow down or halt the progression of the disease. For example, human embryonic stem cells transplanted into PD rat models improve long-term survival and restore motor function [90]. Induced pluripotent stem cell (iPSC)-derived dopaminergic (DA) neurons have the most promising results in preclinical studies, where they successfully induced gradual improvement in motor function after implantation into primate models [91]. In an astrocytic conditioned media, iPSCs cultured with dopaminergic neurons demonstrate active mitochondrial transfer. Using a phopho-p38 pathway, the mitochondria from astrocytes were internalized by the neurons and dopaminergic degenerating neurons were salvaged. When the astrocyte media was ultrafiltered to remove mitochondria, these restorative effects were abolished, suggesting that the primary mechanism of dopaminergic neuronal recovery was due to mitochondrial transfer [73]. These are the preferred line of stem cells for transplantation because they reduce any risk for immunological response. Several clinical trials are ongoing in China, Australia, and Japan to evaluate the efficacy of this treatment.

Amyotrophic Lateral Sclerosis is another neurodegenerative disease heavily influenced by mitochondrial dysfunction. It is largely a sporadic disease without a clear hereditary genetic cause. However, many of the identified genes that do contribute to ALS have roles in impacting mitochondrial function and thus cause the dysfunction of mitochondria as a part of the progression of the disease. For example, mutations in SOD1 and TDP43 in mouse models significantly hinder of mitochondrial transportation and subsequently cause morphological defects and neuronal dysfunction during the early stages of ALS [92]. Sigmar1 mutations, a protein involved in endoplasmic reticulum and mitochondrial communication and calcium balance, also decreases mitochondrial ATP production and impairs neuronal function [68,93]. Using calcium scavenging and inhibiting the stress on the endoplasmic reticulum, mitochondrial dysfunction can be repaired in Sigmar1 mutant mice models of ALS. In addition to restored mitochondrial function, motor neuron degeneration was also diminished [68]. Stem cell therapy for ALS has shifted from attempts to replace lost motor neurons to a goal of providing a neuroprotective local environment for diseased cells [94]. A study examining exosomes from adipose derived stem cells effectively reduced pathological SOD1 levels in mice models of ALS. Furthermore, abnormally phosphorylated CREB and PGC-1α within mitochondria were normalized, proposing that stem cells may be capable of ameliorating ALS pathology via improved mitochondrial function [70]. Studies remain largely in the preclinical or early clinical phase, but results are increasingly promising. For example, the transplantation of human umbilical cord blood cells significantly increases the survival of and delay motor deterioration in SOD1 mouse models [95]. This may be due to the modulation of autoimmune processes to create a neuroprotective environment [96]. Recently, clinical trials aiming to assess the safety of stem cells in humans showed no significantly accelerated deterioration and overall safe clinical outcomes [94]. More studies need to be performed to assess potential benefits to human patients.

## 7. Conclusions

The paucity of effective therapies for stroke alongside our vast knowledge of the mechanisms of secondary cell death following ischemic injury is perplexing. The revolutionary strides in cell-based therapies have revamped treatment goals for stroke. However, the underlying mechanisms of success are frequently contemplated. With a better understanding of how stem cells modulate secondary cell death, these properties can be amplified to strengthen the efficacy of cell-based therapies. As the vitality of functioning mitochondria for stroke recovery is better understood, greater strides in promoting mitochondrial transfer or metabolite exudation from stem cells may significantly magnify their therapeutic benefits.

## Figures and Tables

**Figure 1 antioxidants-12-00178-f001:**
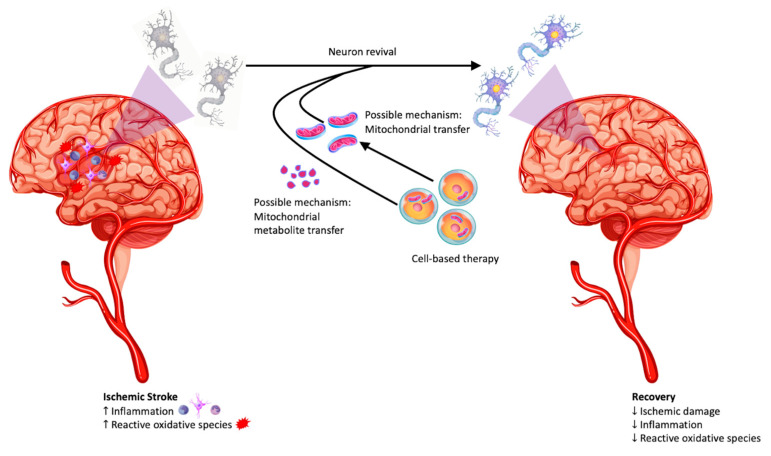
This figure exemplifies the neuroinflammatory and oxidative stress which occurs following an ischemic stroke. Using cell-based therapies, direct mitochondrial transfer theory and mitochondrial metabolite theories are demonstrated here. Mitochondrial restoration promotes recovery by decreasing ischemic damage, inflammation, and ROS.

**Table 1 antioxidants-12-00178-t001:** In Vivo Stem Cell Studies Investigating the Role of Mitochondria in Ischemic Stroke. This table outlines cell-based preclinical trials finding improved functional outcomes attributed to the impact of mitochondrial activity in the background of ischemic insult.

Sample	Cell Type	Route	Dosage	Outcome
MCAO rats	Human Bone Marrow Endothelial Progenitor Cells	Intravenous	4 × 10^6^ cells/mL	Endothelial cells, pericytes, and astrocytes demonstrate near normal morphology without perivascular edema. Mitochondrial morphology in endothelial cells and perivascular astrocytes shows near normal morphology and pinocytic vessels are observed in engrafted cells, which ameliorates post-stroke vasculature [36].
MCAO rats	Miro1-overexpressing Multipotent MSCs	Intravenous	3 × 10^6^ cells/kg	Miro1, normally upregulated in the presence of ROS, promotes mitochondrial transfer to neural cells and reduces neurologic deficits after ischemic stroke [37].
MCAO rats	MSCs	Intra-arterial	5 × 10^5^ cells	Mitochondrial transfer to injured cells of the cerebral microvasculature improves mitochondrial activity, upregulates angiogenesis, improves neurologic function, and decreases infarct volume [38].
MCAO rats	hUC-MSC-derived mitochondria	Intraventricular	Isolate from 3 × 10^7^ cells	Transplanted mitochondria improve ischemic injury exemplified through inhibition of apoptosis, decreased astrogliosis, microglial downregulation, reduced infarct size, and enhanced preservation of motor function [39].
MCAO rats	Ischemic-hypoxic preconditioned olfactory mucosa MSCs	Intravenous	1 × 10^6^ cells	Mitochondrial function is preserved through upregulation of downstream target genes (GRP78 and Bcl-2) by miR-181a and the presence of ROS is significantly reduced. Apoptosis and pyroptosis of neurons are inhibited [40].
Photothrombotic mPFC stroke mice	BM-MSC-derived mitochondria	Intranasal	12 μL	Mitochondrial transplant significantly reduced the presence of ROS in the mPFC following ischemia. Transplant also ameliorates memory impairment, upregulates ATP generation, improves mitochondrial membrane potential, and upregulates expression of synaptic markers (GAP-43 and PSD-95) [41].

**Table 2 antioxidants-12-00178-t002:** In Vitro Stem Cell Studies Investigating the Role of Mitochondria under Oxidative Stress. This table describes in vitro models of oxidative stress pertaining to various elements of mitochondrial function in stroke reperfusion.

Model of Injury	Cell Type	Outcome
Hypoxia-reperfusion	Rat Neural Stem Cells	Coenzyme Q10 achieves an antioxidant effect through interaction with the electron transport chain, increasing expression of survival proteins (pAkt, pGSK3-β, and Bcl-2) and reducing levels of cleaved caspase-3 [42].
OGD	Primary Rat Neural Cells	DJ-1, a protein involved in neuroprotection through regulation of oxidative stress, translocated to mitochondria and enhanced both cell viability mitochondrial activity while reducing ROS concentrations [43].
OGD	Primary Rat Neural Cells	Ischemic conditions promote the uptake of astrocyte-released mitochondrial particles by adjacent neurons, which increases survival signaling [29].
OGD	Human Endothelial Progenitor Cell-derived Extracellular Mitochondria	Incorporation of extracellular mitochondria promotes angiogenesis, decreases BBB permeability, and increases expression of TOM40, mtDNA copy number, and intracellular ATP [44].
Metabolic Switching Paradigm, OGD	Human MSCs, Primary Rat Neurons	Metabolic switching in MSCs yields greater energy generation, respiratory capacity, and ATP production. Co-culture with ODG neurons enhances cellular metabolism, decreases mitochondrial ROS mRNA, and overall improves cell viability [30].
Metabolic Switching Paradigm	Human MSCs	Metabolic switching in MSCs results in mitochondria with enhanced capability for oxidative phosphorylation [25].

**Table 3 antioxidants-12-00178-t003:** In Vitro and In Vivo Studies Targeting Mitochondrial Transfer in Neurodegenerative Diseases. This table delineates the utility of experimental agents to promote mitochondrial transfer in models of neurodegenerative disease.

Sample	Treatment	Route	Dosage	Outcome
Sigmar1(-/-) mice	BAPTA-AM, a selective intracellular calcium chelator or an endoplasmic reticulum stress inhibitor salubrinal	-	-	Restoration of calcium homeostasis and endoplasmic reticulum stability recovered mitochondrial movement and morphology, ultimately reducing motor neuron degeneration [68].
6-hydroxydopamine-induced selective parkinsonian rats	PC12 cell- and Human Osteosarcoma cybrid-derived mitochondria	Intracranial	1.05 μg	Peptide-mediated allogeneic mitochondrial delivery maintains mitochondrial function in the setting of oxidative stress and apoptotic death. Motor activity is improved up to three months following transplantation, and dopaminergic neuron loss is reduced [69].
G93A ALS mice neurons	ADSC-derived exosomes	-	200 μg/mL	Transplanted exosomes reduce aggregation of superoxide dismutase 1 and normalize mitochondrial phospho-CREB/CREB ratio and PGC-1α expression [70].
MPTP-HCL-induced parkinsonian mice	HepG2-derived mitochondria	Intravenous	0.5 mg/kg	Transplanted mitochondria distributed to the brain, liver, kidney, muscle, and heart. PD progression is halted through increased electron transport chain activity, reduced levels of reactive oxygen species, and prevention of apoptosis [71].
Aβ-ICV Alzheimer’s Disease mice	HeLa cell-derived mitochondria	Intravenous	200 μg	Treated mice show enhanced cognitive performance, reduced loss of neurons, and reduced hippocampal gliosis. Mitochondrial function is further ameliorated peripherally in organs such as the liver [72].
Rotenone-induced Parkinson’s neurons	iPSC- and hESC-derived DA neurons and astrocytes	-	-	iPSC-derived astrocytes and astrocytic conditioned media reduce degeneration of dopaminergic neurons and reverse axonal pruning through mitochondrial transfer [73].
5 x Familial Alzheimer’s Disease mice	VBIT-4 (Voltage-Dependent anion channel-1 inhibitor)	PO (in drinking water)	20 mg/kg	VBIT-4 reduces neuronal cell death, downregulates neuroinflammation, and ameliorates metabolic dysfunction, leading to improved cognitive outcomes in behavioral assessments [74].

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
