# Peer review of "Mitochondria in Cell-Based Therapy for Stroke"

_antioxidants, 2023, doi:10.3390/antiox12010178_

Round 1
Reviewer 1 Report
the review by Monsour et al, provides the current developments in treatment and recovery after ischemic stroke. the review article is written well. the authors need to address few of the minor things as below:
1. label the cell types that serves as donor of mitochondria and mitochondrial metabolites. The authors present a complete recovery of the stroke region, that is something that does not happens in real life, please modify the figure accordingly. the figure legend talks about two theories, but only one is presented in figure. either provide both or remove this statement from legend.
2. remove the citation column from tables 1&2 and add the citation at the end of statement in "results" columns. format the table so that the text is not centered in the row. use top alignment. this will improve the visual appeal of the table. Also modify the header "Results" to "Outcome". the "Result" should describe the outcome of the study in brief. remove horizontal lines in between.
3. for table 2, list the name of antioxidant in 1st column, it will help to identify the metabolite and improve the presentation of information.
4. the authors provided a overview of animal studies, it would be more beneficial if current research on similar anti-oxidant studies performed in humans can be provided before the conclusions.
Author Response
Response to Reviewer #1:
- label the cell types that serves as donor of mitochondria and mitochondrial metabolites. The authors present a complete recovery of the stroke region, that is something that does not happens in real life, please modify the figure accordingly. the figure legend talks about two theories, but only one is presented in figure. either provide both or remove this statement from legend.
Response: We have modified the figure and legend to include these important details.
- remove the citation column from tables 1&2 and add the citation at the end of statement in "results" columns. format the table so that the text is not centered in the row. use top alignment. this will improve the visual appeal of the table. Also modify the header "Results" to "Outcome". the "Result" should describe the outcome of the study in brief. remove horizontal lines in between.
Response: Thank you for the suggestions, we agree that such modifications will enhance the visual appeal of the table and to this end have implemented the changes.
- for table 2, list the name of antioxidant in 1st column, it will help to identify the metabolite and improve the presentation of information.
Response: While we strongly considered adding a column to clearly delineate the primary metabolite, it was ultimately decided that this was better left described in the “outcomes” column, as the majority of studies included in the table did not identify a single metabolite of interest in the antioxidant response.
- the authors provided a overview of animal studies, it would be more beneficial if current research on similar anti-oxidant studies performed in humans can be provided before the conclusions.
Response: We have added the following:
A number of clinical trials have been conducted with the goal of reducing oxidative stress following stroke [27]. Lipoic acid, involved in vitamin C and E recycling, is being tested in an ongoing clinical trial on reperfusion therapies for ischemic stroke in patients with diabetes (NCT 04041167). Salvianolic acid is also being studied as an anti-oxidant following stroke, as it is a potent oxidation, coagulation, and platelet aggregation inhibitor (NCT04931628). Edaravone, a ROS scavenger, has shown reduced mortality when ad-ministered with alteplase therapy or after endovascular revascularization [59,60]. Importantly, not all preclinical successes are translatable to clinical settings, however, and further investigation is needed [27,61-67].
Reviewer 2 Report
This manuscript describes concisely the current situation of stem cell therapy of neurodegenerative diseases. It contributes without doubt to further development of this area.
Especially mitochondrial transfer from stem cells to astrocytes or other types of cells is a novel concept in this area and one of the scientific merits of this manuscript, Having said that, the authors should pay attention to the following points. One of the devastating aspects of ischemic stroke is known to be reperfusion injury, which, of course, causes oxidative stress. It may be more comfortable for the readers to see these words in the text, although it does not change any essential points of the article.
Author Response
Response to Reviewer #2:
- One of the devastating aspects of ischemic stroke is known to be reperfusion injury, which, of course, causes oxidative stress. It may be more comfortable for the readers to see these words in the text, although it does not change any essential points of the article.
Response: We agree that this is important to include to ease the reading and have added the following: “Furthermore, reperfusion following ischemic stroke generates a large number of ROS contributing to oxidative stress [27].” We also changed the subtitle: 2. Mitochondrial impairment in the oxidative stress following stroke and reperfusion injury”
Reviewer 3 Report
The manuscript is a comprehensive literature review on the role of mitochondria in the pathogenesis of stroke and their possible implication in cell-based therapies in pathologies accompanied by oxidative stress.
The thematic sections of the manuscript are adequately set focusing on: mitochondrial impairment in the oxidative stress following stroke; the repair of damaged mitochondria in stroke: the astrocytes - to neurons transfer of mitochondria; the stem cell-neural cell crosstalk and stem cell-based mitochondrial repair in disorders of the CNS.
All sections are thoroughly elaborated based on recent original publications and concepts like the direct mitochondrial transfer and mitochondrial metabolite theories.
I would suggest in the last section examining the application of stem cell-based mitochondrial repair in degenerative CNS disorders such as Alzheimer's disease, Parkinson’s disease, and amyotrophic lateral sclerosis to be added a paragraph on autistic spectrum disorder (ASD) as it is also accompanied by neurodegeneration and mitochondrial dysfunction. The title of the manuscript could be edited (in case the authors agree) as just 'Mitochondria in Cell-based Therapy for Stroke'.
The manuscript presents a current topic and offers new perspectives for cell-based therapies in brain hypoxic conditions focusing predominantly on stroke. It has high informative and scientific merit and could be accepted for publication in Antioxidants.
Author Response
Response to Reviewer #3:
- I would suggest in the last section examining the application of stem cell-based mitochondrial repair in degenerative CNS disorders such as Alzheimer's disease, Parkinson’s disease, and amyotrophic lateral sclerosis to be added a paragraph on autistic spectrum disorder (ASD) as it is also accompanied by neurodegeneration and mitochondrial dysfunction.
Response: We appreciate your thorough and thoughtful feedback on our manuscript. After review of the literature regarding mitochondrial transfer and stem cell therapies for ASD, we agree that this is an emerging body of literature given the mitochondrial dysfunction seen in ASD. Given the current ethical qualms regarding discussion on “treating” ASD, we hope you can understand our decision to not add this section [1].
- The title of the manuscript could be edited (in case the authors agree) as just 'Mitochondria in Cell-based Therapy for Stroke'.
Response: We agree that this title is more informative and straight forward given our paper’s topic and have adjusted it accordingly.